# Evaluating the frequency of English language requirements in clinical trial eligibility criteria: A systematic analysis using ClinicalTrials.gov

**Akila V. Muthukumar[1], Walker Morrell[2,3], Barbara E. Bierer** [2,3,4] *

**1** Harvard College, Cambridge, Massachusetts, United States of America, **2** Multi-Regional Clinical Trials Center of Brigham and Women's Hospital and Harvard, Boston, Massachusetts, United States of America, **3** Division of Global Health Equity, Department of Medicine, Brigham and Women's Hospital, Boston, Massachusetts, United States of America, **4** Department of Medicine, Harvard Medical School, Boston, Massachusetts, United States of America

* bbierer@bwh.harvard.edu

**Data Availability Statement:** All relevant data are within or cited in the manuscript or available from ClinicalTrials.gov.

## Abstract

### Background

A number of prior studies have demonstrated that research participants with limited English proficiency in the United States are routinely excluded from clinical trial participation. Systematic exclusion through study eligibility criteria that require trial participants to be able to speak, read, and/or understand English affects access to clinical trials and scientific generalizability. We sought to establish the frequency with which English language proficiency is required and, conversely, when non-English languages are affirmatively accommodated in US interventional clinical trials for adult populations.

### Methods and findings

We used the advanced search function on ClinicalTrials.gov specifying interventional studies for adults with at least 1 site in the US. In addition, we used these search criteria to find studies with an available posted protocol. A computer program was written to search for evidence of English or Spanish language requirements, or the posted protocol, when available, was manually read for these language requirements. Of the 14,367 clinical trials registered on ClinicalTrials.gov between 1 January 2019 and 1 December 2020 that met baseline search criteria, 18.98% (95% CI 18.34%–19.62%; $n = 2,727$) required the ability to read, speak, and/or understand English, and 2.71% (95% CI 2.45%–2.98%; $n = 390$) specifically mentioned accommodation of translation to another language. The remaining trials in this analysis and the following sub-analyses did not mention English language requirements or accommodation of languages other than English. Of 2,585 federally funded clinical trials, 28.86% (95% CI 27.11%–30.61%; $n = 746$) required English language proficiency and 4.68% (95% CI 3.87%–5.50%; $n = 121$) specified accommodation of other languages; of the 5,286 industry-funded trials, 5.30% (95% CI 4.69%–5.90%; $n = 280$) required English and 0.49% (95% CI 0.30%–0.69%; $n = 26$) accommodated other languages. Trials related to infectious disease were less likely to specify an English requirement than all registered trials

**Funding:** The authors received no specific funding for this work.

**Competing interests:** The authors have declared that no competing interests exist.

**Abbreviations:** RR, relative risk.

(10.07% versus 18.98%; relative risk [RR] = 0.53; 95% CI 0.44–0.64; $p < 0.001$). Trials related to COVID-19 were also less likely to specify an English requirement than all registered trials (8.18% versus 18.98%; RR = 0.43; 95% CI 0.33–0.56; $p < 0.001$). Trials with a posted protocol ($n = 366$) were more likely than all registered clinical trials to specify an English requirement (36.89% versus 18.98%; RR = 1.94, 95% CI 1.69–2.23; $p < 0.001$). A separate analysis of studies with posted protocols in 4 therapeutic areas (depression, diabetes, breast cancer, and prostate cancer) demonstrated that clinical trials related to depression were the most likely to require English (52.24%; 95% CI 40.28%–64.20%). One limitation of this study is that the computer program only searched for the terms "English" and "Spanish" and may have missed evidence of other language accommodations. Another limitation is that we did not differentiate between requirements to read English, speak English, understand English, and be a native English speaker; we grouped these requirements together in the category of English language requirements.

## Conclusions

A meaningful percentage of US interventional clinical trials for adults exclude individuals who cannot read, speak, and/or understand English, or are not native English speakers. To advance more inclusive and generalizable research, funders, sponsors, institutions, investigators, institutional review boards, and others should prioritize translating study materials and eliminate language requirements unless justified either scientifically or ethically.

### Author summary

#### Why was this study done?

- Some clinical trials in the US exclude individuals who do not read, speak, or write English.

- While requiring English language proficiency for entry into clinical trials may sometimes be scientifically or ethically justified, often it is not, raising concerns of equity and justice.

- We sought to establish the frequency of English language requirements in clinical trials in the United States.

#### What did the researchers do and find?

- We reviewed 14,367 US clinical trials registered on ClinicalTrials.gov in 2019 and 2020.

- Of the 14,367 clinical trials, 18.98% ($n = 2,727$) had English language requirements, and 2.71% ($n = 390$) mentioned accommodation of a language other than English.

- Clinical trials funded by the federal government were more likely to require English than clinical trials funded by the life sciences and pharmaceutical industries.

- Compared to all clinical trials in 2019 and 2020, clinical trials related to COVID-19 and other infectious diseases were less likely to have English language requirements.

- In a separate analysis of clinical trials in 4 therapeutic areas (depression, diabetes, breast cancer, and prostate cancer), trials related to depression were the most likely to require English proficiency (52.24%).

### What do these findings mean?

- A meaningful percentage of clinical trials in the US exclude individuals who do not speak, read, or write English.
- Clinical trials can be made more inclusive by eliminating language requirements that are not scientifically or ethically justified.

## Introduction

The populations enrolled in clinical trials should optimally reflect the characteristics of the populations for whom the knowledge gained from the research is applicable [1]. While restricting clinical trial participation for scientific or ethical reasons, such as safety concerns or protection of vulnerable populations, is justified, the intentional or systematic exclusion of subgroups of the population for whom the research is intended is problematic. Scientifically, such exclusion limits the generalizability of the research findings, resulting, inter alia, in an insufficient evidence base for the safe and effective use of medicines and other interventions. Such exclusion is also ethically problematic in that, as a principle of justice, the selection of research participants is not equitable; exclusion may also contribute to distrust of the clinical research enterprise and healthcare systems, institutions, and providers. Finally, the exclusion contributes to health inequities, a problem that is particularly salient when applied to individuals prohibited from participating simply for their lack of English proficiency in the absence of other reasons.

Requiring English proficiency in a clinical trial may be scientifically justified, such as when a trial involves surveys, assessments, or outcome measures that have been validated only in English, at least in an effort not to delay the initiation of research due to translation and validation. Research evaluating or using mobile health technologies available only in English will also require English proficiency. However, no such scientific justification exists for the exclusion of non-English speakers from clinical trials for pragmatic reasons such as the cost of translation, or inadequate training or availability of language-concordant staff or interpreters [2]. And in interventional therapeutic trials, where there is a possibility of direct benefit to the participant, the exclusion of individuals based on language alone is itself inequitable.

A number of prior studies have demonstrated that research participants with limited English proficiency in the United States are routinely excluded from clinical trial participation; eligibility criteria require that trial participants be able to speak, read, and/or understand English [3–5]. These studies have routinely called for a reexamination of this requirement and the translation of study materials to different languages. The term "limited English proficiency" is used here to describe individuals who have a limited ability to read, speak, write, or understand English. In the US, an estimated 57 million people do not speak English, and an additional 25 million people may be defined as having limited English proficiency [6].

We sought to establish the frequency with which English proficiency is required in clinical trials in the US. Using the eligibility criteria of clinical trials registered on ClinicalTrials.gov,

and posted study protocols when available, we evaluated the frequency of language require-ments in eligibility criteria, and we further characterized the requirement by funding source and across several therapeutic areas. We also reviewed the frequency with which non-English languages are affirmatively accommodated in clinical trials in the US. We discuss the common —and discriminatory—exclusion of individuals who do not speak, read, or understand English from clinical trials in the US and the ethical challenges raised by this exclusion.

## Methods

We used the advanced search function on ClinicalTrials.gov to search for interventional stud-ies for adults (18–64 years) and older adults (65+ years) with at least 1 site in the US with an actual start date between 1 January 2019 and 1 December 2020. We performed additional searches for interventional studies for adults (18–64 years) and older adults (65+ years) with at least 1 site in the US across 4 therapeutic areas (depression, diabetes, breast cancer, and pros-tate cancer). We restricted these additional searches to studies that had posted a study protocol to ClinicalTrials.gov, and searched over a 4-year time interval, or longer if needed, so that the search results contained at least 50 trials. The criteria for each search can be found in Table A in S1 Tables.

The comma-separated values (CSV) files of the search results were downloaded. The fund-ing source data element for each trial was coded as "federal," "industry," "federal/industry," or "other" using the categories outlined in Table B in S1 Tables.

Data analysis followed a prospective analysis plan. We wrote a computer program in R to parse each trial's ClinicalTrials.gov webpage entry for the terms "English" and "Spanish" for all trials with an actual start date between 1 January 2019 and 1 December 2020. If the program identified either term on a trial's ClinicalTrials.gov webpage entry, it printed the line of text in which the term was found.

One person (AM or WM) manually reviewed the printed text and categorized each trial as one that (1) requires English, (2) accommodates translation, (3) mentions that participants must be able to provide informed consent and/or communicate with study staff via phone call or email, or (4) does not mention any language requirement for participation ("no mention"). After reviewing a sample of clinical trials, we determined that a statement about the ability to provide informed consent is a common inclusion criterion and independent of language. Since there was no relevant distinction between categories 3 and 4, these 2 categories were pooled for further analysis. Further, since Spanish is the second most common language in the US, we noted the number of trials that specifically mentioned the accommodation of Spanish as a subcategory of category 2.

Details on the specific entries that informed language categorization are provided in Table 1.

If the terms "English" or, for accommodation of language translation, "Spanish" were not found by the computer program and a protocol was posted, one person categorized the trial based on the study protocol. If no protocol was posted, the trial was categorized as "no men-tion." This method was chosen so that findings of English language requirements were conser-vative and not overestimated.

To verify reliability of the computer program, we manually reviewed all trials in 1 therapeu-tic area (diabetes, $n = 85$). We found 100% agreement with the results generated by the com-puter program. The computer program is available at https://github.com/akilamuthukumar/languagesearchfunctions and https://rpubs.com/akilamuthukumar/709377. The data are avail-able upon request.

Table 1. Key phrases used to code language requirements in clinical trials.

| Classification | ClinicalTrials.gov entry |
|---|---|
| Requires English | Native English speakers |
| | Ability to read English |
| | Ability to speak English |
| | Ability to understand English |
| | Legally authorized representative must read, speak, understand English |
| | We have no non-English speaking patients in this population |
| Accommodates translation | Mentions another language by name: Spanish, American Sign Language, Korean, Xhosa, Swahili, Luganda, French, Arabic, Mandarin, Cantonese, etc. |
| | Non-English speakers will be accommodated |
| | If applicable, [informed consent] will be provided in a certified translation of the local language |
| | Non-English language speaking participants for whom an Institutional Review Board (IRB) approved short form is available |
| | The informed consent form must be written in a language fully comprehensible to the prospective patient |
| | Any oral or written information will be provided to participants in their own language |
| | Patients unable to read/write English are eligible to participate in the overall study but will not be required to participate in the Patient-Reported Outcome questionnaires |
| | Self-reported questionnaires will be administered in countries where the questionnaires have been translated into the native language of the region and linguistically validated |
| | Standard Operating Procedures (SOPs) are in place pertaining to how to approach and consent participants of limited English proficiency |

Four therapeutic areas (depression, diabetes, breast cancer, and prostate cancer) for which study protocols had been uploaded to ClinicalTrials.gov were further examined. Two independent coders (AM and WM) manually reviewed and classified the ClinicalTrials.gov webpage entry for each trial. If language requirements were not identified on a trial's webpage entry, the 2 coders reviewed the trial protocol and categorized the trial as one that (1) requires English, (2) accommodates translation, or (3) does not mention any language requirement for participation. The inter-rater reliability was 95.3%, and entry errors ($n = 4$) were corrected. Discrepancies ($n = 9$) were reviewed by an independent third party (BEB) and discussed until consensus was reached.

The frequencies of language requirements were determined, and 95% confidence intervals were calculated. Comparisons were made between proportions of trials by calculating the relative risk (RR). 95% confidence intervals for these comparisons were calculated on the natural log scale and converted to a linear scale through exponentiation. $p$-Values were calculated using a 2-tailed $z$-test of 2 proportions at significance level 0.001.

In order to determine whether there was a relationship between the accommodation of a non-English language and increased diverse representation in clinical trials, we reviewed the ethnicity data in the trials with an actual start date between 1 January 2019 and 1 December 2020 that had posted results to ClinicalTrials.gov.

## Results

In total, 14,367 interventional clinical trials met the advanced search criteria of having at least 1 site in the US, being intended for adults/older adults, and being registered on ClinicalTrials.gov between 1 January 2019 and 1 December 2020 (Table 2). Of these clinical trials, having the ability to read, speak, and/or understand English, or being a native English speaker, was required by 18.98% (95% CI 18.34%–19.62%; $n = 2,727$), while 2.71% (95% CI 2.45%–2.98%;

**Table 2. Language requirements in clinical trials.**

| Trial type | Total | Number (percent, 95% CI) | | | |
| --- | --- | --- | --- | --- | --- |
| | | Requires English | Accommodates translation | Accommodates Spanish | Does not mention any language requirement |
| All trials | 14,367 | 2,727 (18.98, 18.34–19.62) | 390 (2.71, 2.45–2.98) | 328 (2.28, 2.04–2.53) | 11,250 (78.30, 77.63–78.98) |
| Infectious disease trials | 1,023 | 103 (10.07, 8.22–11.91) | 32 (3.13, 2.06–4.19) | 31 (3.03, 1.98–4.08) | 888 (86.80, 84.73–88.88) |
| COVID-19 trials | 611 | 50 (8.18, 6.01–10.36) | 15 (2.45, 1.23–3.68) | 15 (2.45, 1.23–3.68) | 546 (89.36, 86.92–91.81) |
| Trials with posted protocol | 366 | 135 (36.89, 31.94–41.83) | 18 (4.92, 2.70–7.13) | 15 (4.10, 2.07–6.13) | 213 (58.20, 53.14–63.25) |
| Federally funded trials | 2,585 | 746 (28.86, 27.11–30.61) | 121 (4.68, 3.87–5.50) | 105 (4.06, 3.30–4.82) | 1,718 (66.46, 64.64–68.28) |
| Industry funded trials | 5,286 | 280 (5.30, 4.69–5.90) | 26 (0.49, 0.30–0.69) | 20 (0.38, 0.21–0.54) | 4,980 (94.21, 93.58–94.84) |

The 95% confidence intervals are reported as percentages of trials that meet the cell characteristic.

$n$ = 390) specifically mentioned accommodation of translation to another language. The majority (78.3%; 95% CI 77.63%–78.98%; $n$ = 11,250) did not mention any language requirement.

Of the 14,367 trials, 366 (2.55%) had posted a study protocol. Notably, these trials with a protocol were more likely than all registered clinical trials to specify an English requirement (36.89% versus 18.98%; RR = 1.94, 95% CI 1.69–2.23; $p < 0.001$). Of the 366 trials, 11.8% (95% CI 8.21%–14.74%; $n$ = 42) had information about English language requirements available only in the posted study protocol and not on the trial's ClinicalTrials.gov webpage entry.

## Differences in language requirements by funding source

The relationship between funding source and language requirement was examined. Trials that were co-funded by both federal and industry sources (e.g., industry/National Institutes of Health) were excluded from the analysis by funding source. Federally funded trials were more likely to specify an English requirement than industry-funded trials (28.86% versus 5.30%; RR = 5.45; 95% CI 4.79–6.20; $p < 0.001$). Federally funded trials were also more likely to specify accommodation of translation than industry-funded trials (4.68% versus 0.49%; RR = 9.52; 95% CI 6.18–14.65; $p < 0.001$) (Table 2).

## Differences in language requirements by therapeutic area

We hypothesized that interventional studies addressing the COVID-19 pandemic may be more accommodating of other languages (either by not requiring English proficiency or specifically mentioning additional translation, such as Spanish) given the disproportionate prevalence of infection in Hispanic and Latino populations in the US [7,8]. Of the 14,367 clinical trials we reviewed across 2019 and 2020, trials related to infectious diseases ($n$ = 1,023) of which the majority (59.7%) were related to COVID-19 ($n$ = 611), were less likely to specify an English requirement than all registered trials (trials related to infectious diseases versus all trials: 10.07% versus 18.98%; RR = 0.53; 95% CI 0.44–0.64; $p < 0.001$; trials related to COVID-19 versus all trials: 8.18% versus 18.98%; RR = 0.43; 95% CI 0.33–0.56; $p < 0.001$).

We observed that reviewing the eligibility criteria of posted study protocols was a more accurate method of determining language requirements than reviewing the selected criteria available on a trial's ClinicalTrials.gov webpage entry. In order to examine whether there were differences in other therapeutic areas, we reviewed registered trials with an available study protocol over a 4-year time interval, or longer if needed, to obtain at least 50 trials. There were

**Table 3. Language requirements in clinical trials by therapeutic area.**

| Therapeutic area | Total | Number (percent, 95% CI) | | | |
|---|---|---|---|---|---|
| | | Requires English | Accommodates translation | Accommodates Spanish* | Does not mention any language requirement |
| Depression | 67 | 35 (52.24, 40.28–64.20) | 5 (7.46, 1.17–13.76) | 0 (NA) | 27 (40.30, 28.55–52.04) |
| Diabetes | 85 | 25 (29.41, 19.73–39.10) | 13 (15.29, 7.64–22.95) | 8 (9.41, 3.20–15.62) | 47 (55.29, 44.72–65.86) |
| Breast cancer | 70 | 13 (18.57, 9.46–27.68) | 15 (21.43, 11.82–31.04) | 5 (7.14, 1.11–13.18) | 42 (60.00, 48.52–71.48) |
| Prostate cancer | 55 | 4 (7.27, 0.41–14.14) | 8 (14.55, 5.23–23.86) | 0 (NA) | 43 (78.18, 67.27–89.10) |

*Number of trials specifying Spanish is a subset of the trials accommodating translation.

NA, not applicable.

clear differences in the requirement for English proficiency by therapeutic area: Requirements in trials related to depression (35/67; 95% CI 40.28%–64.20%) exceeded those related to diabetes (25/85; 95% CI 19.73%–39.10%), breast cancer (13/70; 95% CI 9.46%–27.68%), and prostate cancer (4/55; 95% CI 0.41%–14.14%) (Table 3). Mandating English as a requirement did not correlate with likelihood of accommodating translation, at least in this small sample.

## Non-English-language accommodation and clinical trial enrollment

To determine whether accommodation of non-English language correlated with an increase in the ethnicities of research participants, we reviewed trials with posted results. Of the 14,367 clinical trials initiated in 2019 and 2020, only 323 had posted results on ClinicalTrials.gov. Of the 323 trials that had posted results, only 189 reported ethnicity data, and of those 189 trials, only 4 were identified as specifically mentioning accommodation of a non-English language. The small sample size of completed studies with posted results impeded any correlative analyses on non-English-language accommodation and the ethnicities of research participants.

## Discussion

While many trials do not include a language requirement as an eligibility criterion, this study found that a substantial percentage of interventional clinical trials for adults in the US required that participants be native English speakers or be able to speak, read, and/or understand English. Exclusion from participation on the basis of language varied by therapeutic area and by funding source.

The specific eligibility criteria describing the English language requirements in clinical trials varied in wording; the intention and impact of that variability are uncertain. Whether "native English speaker" requirements differentiate between native English speakers and non-native English speakers who are nevertheless proficient in English, for instance, and whether research staff actually distinguish between the 2 is unclear. Similarly, the scientific reasons for clinical trial requirements for participants to be able to speak, read, or understand English, or possess the capacity for all 3, were neither explicit nor evident. Further, very few protocols included methods for determining English proficiency. How these requirements are applied by study staff, therefore, is subject to interpretation.

In comparison to the proportion of clinical trials that specified English requirements, fewer clinical trials explicitly accommodated translation into languages other than English. It is possible that this number is underestimated since trials that did not specify translation on ClinicalTrials.gov may have been registered on other clinical trial registries in non-English languages, trial protocols were not always available for review, and the computer program only searched for the terms "English" and "Spanish." While the finding of an affirmative

requirement for English proficiency, therefore, is of greater reliability, the infrequency with which translation was mentioned, even in interventional trials addressing the COVID-19 pandemic, was notable. As a related point, it should not be assumed that clinical trials categorized as "accommodates translation" are sufficiently inclusive. Any trial that accommodated at least 1 other language was categorized as "accommodates translation," but just and appropriate inclusion of diverse participants often requires accommodation of more than a single additional language.

The information available on ClinicalTrials.gov leads to an underestimation of the language requirements of clinical trial eligibility compared to posted study protocols. The responsible party for registering the clinical trial has latitude in which eligibility criteria they choose to include on the trial's ClinicalTrials.gov webpage entry [9]; neither the ClinicalTrials.gov Protocol Registration and Results System [10] nor Section 801 of the Food and Drug Administration Amendments Act of 2007 (FDAAA 801) [11] requires language proficiency to be an essential element of registration. The finding that the subset of clinical trials with a posted protocol were more likely to specify language requirements as part of the eligibility criteria compared to all clinical trials suggests that some responsible parties do not consider language requirements to be essential information. We recommend that responsible parties include all eligibility criteria, including language requirements, on ClinicalTrials.gov, as this information is important for clinicians, patients, families, and others to identify clinical trial opportunities. If all eligibility criteria are not included in the trial's ClinicalTrials.gov website entry, then responsible parties should, at a minimum, upload the trial protocol containing such information to ClinicalTrials.gov.

It is surprising that federally funded trials were not only more likely to require English but also more likely to accommodate translation than industry-funded trials. Industry-funded trials are often better resourced and more multinational than federally funded trials, and therefore translation may be less likely to be mentioned. In contrast, when federally funded trials have the capacity for translation, the investigators may be more likely to make this capacity explicit.

Of the therapeutic areas analyzed, clinical trials for depression were the most likely to require English proficiency (52.24%). Whether this disproportionate requirement for English proficiency is related to the necessity of language concordance between participant and investigator, the limited availability of validated data collection and survey instruments in non-English languages, the more qualitative aspect of this research, or another reason is not clear; study protocols routinely failed to provide an explanation. It is noteworthy that trials for both breast and prostate cancer were less likely than trials for depression or diabetes to require English proficiency, a difference potentially related to cancer outcome measures being derived from imaging or laboratory studies, as well as to a focus on inclusion of diverse populations in oncology [12]. However, these results are based on small numbers and subject to variation; additional research is needed. That interventional studies addressing infectious diseases in general, and COVID-19 specifically, less often required English proficiency compared to all clinical trials is reassuring, given the prevalence and severity in Hispanic and Latino populations.

There are 2 limitations to the study methodology. One limitation is that the computer program only searched for the terms "English" and "Spanish"; if neither of these terms was identified, and a protocol was not available for manual review, language requirements that did not use the terms "English" or "Spanish" may have been overlooked. Another limitation is that the requirements to read, speak, and/or understand English and/or be a native English speaker were grouped together in the category of English language requirements.

Our findings highlight the prevalence of the routine exclusion of adult non-English-speaking individuals from interventional clinical trials with at least 1 site in the US. In some cases, exclusion may be justified for scientific or methodological reasons, such as the unavailability of a validated data collection tool in languages other than English; one would hope, however, that barriers of this nature would be temporary and other tools would be identified or developed. In some cases, language concordance between participant and investigator or research staff may be necessary, although whether that need is scientific or a matter of convenience and cost is unclear. Clinical care (including psychiatric care) is able to cross a language divide through translation, interpreters, and other means; clinical research should be held to the same expectation. In addition to depriving individuals of access to clinical trials and, most acutely, to those trials with potential therapeutic benefit, lack of appropriate representation in clinical trials limits the generalizability of the results of the research.

Funders and sponsors, institutions, investigators and their study teams, institutional review boards, and other stakeholders have the capacity to redress the problem. Funders and sponsors could include translation as an allowable cost in grant applications or contracts or provide translated documents as a matter of course. A short-form consent document summarizing the basic elements of informed consent in a non-English language is sometimes used to document that the elements of informed consent were presented orally, but such a document does not offer study-specific information. This approach may be used, but optimally should be a temporary fix. Future work should delineate the conditions under which use of a short-form consent document is adequate and the conditions under which translation of the informed consent and other study materials is necessary or expected. Institutions and investigators with access to translation services for clinical purposes could extend the services to research, and provision of interpreter and translation resources might incentivize inclusion of diverse populations. Institutional review boards could develop standardized guidance for investigators and their study teams since such guidance does not routinely exist currently [13,14]. These strategies and others, coupled with appropriate resources, will help advance more inclusive and generalizable research.

## Supporting information

**S1 Tables.** Table A: ClinicalTrials.gov advanced search criteria. Table B: ClinicalTrials.gov funder type classification.
(DOCX)

## Acknowledgments

We thank Graeme Peterson for assistance with the statistical techniques used in this paper.

## Author Contributions

**Conceptualization:** Barbara E. Bierer.

**Data curation:** Akila V. Muthukumar, Walker Morrell, Barbara E. Bierer.

**Formal analysis:** Akila V. Muthukumar, Walker Morrell, Barbara E. Bierer.

**Investigation:** Akila V. Muthukumar, Walker Morrell, Barbara E. Bierer.

**Methodology:** Akila V. Muthukumar, Barbara E. Bierer.

**Project administration:** Barbara E. Bierer.

**Resources:** Barbara E. Bierer.

**Software:** Akila V. Muthukumar.

**Supervision:** Barbara E. Bierer.

**Validation:** Akila V. Muthukumar, Walker Morrell.

**Writing – original draft:** Akila V. Muthukumar, Walker Morrell, Barbara E. Bierer.

**Writing – review & editing:** Akila V. Muthukumar, Walker Morrell, Barbara E. Bierer.

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
