## [Editor Report · Decision Letter 0]

5 Apr 2021

Dear Dr Bierer, 

Thank you for submitting your manuscript entitled "English Proficiency Requirements in Clinical Trial Eligibility: Common, Persistent, and Inequitable" for consideration by PLOS Medicine.

Your manuscript has now been evaluated by the PLOS Medicine editorial staff and I am writing to let you know that we would like to send your submission out for external peer review.

Kind regards,

Beryne Odeny

Associate Editor

PLOS Medicine

---

## [Decision Letter · Decision Letter 1]

11 May 2021

Dear Dr. Bierer,

Thank you very much for submitting your manuscript "English Proficiency Requirements in Clinical Trial Eligibility: Common, Persistent, and Inequitable" (PMEDICINE-D-21-01563R1) for consideration at PLOS Medicine. 

[LINK]

In light of these reviews, I am afraid that we will not be able to accept the manuscript for publication in the journal in its current form, but we would like to consider a revised version that addresses the reviewers' and editors' comments. Obviously we cannot make any decision about publication until we have seen the revised manuscript and your response, and we plan to seek re-review by one or more of the reviewers. 

We expect to receive your revised manuscript by Jun 01 2021 11:59PM. Please email us (plosmedicine@plos.org) if you have any questions or concerns.

We look forward to receiving your revised manuscript. 

Sincerely,

Beryne Odeny, 

PLOS Medicine 

plosmedicine.org

Thank you for your submission. Before we proceed, please address the following editorial and reviewer comments.

1) Please revise your title according to PLOS Medicine's style. Your title must be nondeclarative and not a question. It should begin with main concept if possible. Please place the study design (e.g. "A cross-sectional study" or “systematic review”) in the subtitle (i.e., after a colon). 

2) Abstract summary - At this stage, we ask that you reformat your non-technical Author Summary. The Author Summary should immediately follow the Abstract in your revised manuscript. This text is subject to editorial change and should be distinct from the scientific abstract. The summary should be accessible to a wide audience that includes both scientists and non-scientists. Please see our author guidelines for more information: https://journals.plos.org/plosmedicine/s/revising-your-manuscript#loc-author-summary.

3) In the abstract Methods and Findings:

a) Please revise the subheading “Methods/Finding” to “Methods and Findings,”

b) Please ensure that all numbers presented in the abstract are present and identical to numbers presented in the main manuscript text.

c) Please quantify the main results with both 95% CIs and p values.

d) In the last sentence of the Abstract Methods and Findings section, please describe the main limitation(s) of the study's methodology.

4) Did your study have a prospective protocol or analysis plan? Please state this (either way) early in the Methods section.

5) In the Methods and Results section:

a) Please provide 95% CIs and p values for estimates in the main text and tables

b) When a p value is given, please specify the statistical test used to determine it.

c) Please do not report P<0.01; report as P < 0.001.

6) Figures and tables:

a) The presentation in table 3 is confusing. Please remove “n(%)” from the rows and instead write it at the top of the last 2 columns(“Requires English” and “Accommodates translation”)

b) Please define the following abbreviations in your tables 1: ICF, ASL. 

c) In table 1, please replace "subject" with participant, patient, individual, or person.

7) Please look separately at Spanish in the studies since that is the second most frequently spoken language and note how often that is mentioned/accommodated

8) Please use the "Vancouver" style for reference formatting and see our website for other reference guidelines https://journals.plos.org/plosmedicine/s/submission-guidelines#loc-references. . Please ensure that weblinks are current and accessible

Comments from the reviewers:

Reviewer #1: I confine my remarks to statistical aspects of this paper.

While what was done isn't really wrong, I don't think it's ideal I am concerned that over three-quarters of the studies did not mention translation. This is a kind of missing data problem, but the usual methods of multiple imputation are likely to fail, since so much data was missing. Therefore, rather than use z-tests of two proportions, I would a) Simply look at the proportion that do accommodate translation and b) use multinomial logistic regression to examine whether the variable "translation" (with three levels) varies by any other trait you are interested in. This treats "missing" as its own category, which is, I think, a better approach than simply ignoring them.

Peter Flom 

Reviewer #2: This is a well-written and interesting paper that addresses an important issue in clinical trial eligibility. Strengths of the paper include a thoughtful introduction highlighting the importance of removing barriers to research participation to increase diversity in clinical trials and the use of the use of a rigorous and reproducible search strategy to identify language requirements for clinical trials with at least one US site. The main findings include 19.0% of trials on ClinicalTrials.gov specifying an English language requirement and only 2.7% specifically mentioning accommodation of translation to another language. Compared to posted protocols, the information in ClinicalTrials.gov under-reported both English language requirements and accommodations for translation. There are a number of things that could be done to further strengthen the paper, both in terms of the results presented and the discussion.

Major Comments:

1) It would be particularly helpful, and a greater contribution to the literature, if the authors could demonstrate a relationship between accommodation for translation and actual increased diversity in clinical trials. While this would not be possible for all languages, it seems like it should be possible to search information on ClinicalTrials.gov to determine whether accommodation of translation resulted in overall greater proportions of non-white participants. Another approach would be to examine the relation between accommodation of Spanish translation and the proportion of Hispanic participants. This information would provide the missing link in the mechanism linking translation to participation.

2) Among the possible accommodations for translated documentation, it seems like a missed opportunity not to comment on the potential for using the consent short form (21 CFR 50.27).

3) An additional important point for inclusion of non-English speaking participants in trials is to modify the incentives and disincentives for sites to include these participants. These include covering not only the translation costs but the costs associated with use of interpreter services and of added staff time required for visits conducted using an interpreter.

4) There should be at least mention of the distinction between clinical trial sites potentially being able to offer staff fluent in some languages (e.g. Spanish) and clinical trial sites with truly broader language capacity (e.g. through interpreter services), which could accommodate a much wider range of languages. Stated another way, the way this analysis is presented, all trials would be 'inclusive' if they just included Spanish - which may be easier to achieve. However, true equity in access to trials will require moving beyond a single additional language.

Reviewer #3: This is a timely and important topic, and I commend the authors for submitting it to a systematic analysis. It is to be hoped that studies such as this spark a trend that will bring more equitable representation in clinical research. Below are specific comments:

Abstract: the percentages under the results section do not add up to 100, clarify that the rest of the percentage were not eligible for inclusion, did not specify language, etc.

Are the phrases "ability to read, speak, and/or understand English", "English language proficiency", "required English", "required English language", and "mandate English" interchangeable? Similarly, "who cannot read, speak, and/or understand English" and "are non-Native English speakers" can mean widely different things. Consistent terms should be used in the abstract. A definition of Limited English Proficiency (LEP) is offered in the introduction

Perhaps the larger issue is that protocols do not have a consistent language to classify proficiency and that is also something that needs to change (this point is made on page 8, but the abstract uses the terms interchangeably).

Page 2, Line 37-38 belongs in the conclusion (not introduction)

Page 2, Line 41-42 - what does "varying time points" mean?

Page 6, Line 111: The term Latinx has gained popularity in academic circles, but it is less known, widespread and accepted in non-academic communities in the United States that might be expected to identify as "Latinx" (very few people actually do). It is especially important to select terms carefully or perhaps acknowledge the complexities of using rarified academic language in an article concerned with increasing representation.

Page 11, Lines 201-208: The last paragraph outlines some general suggestions, but it would be a stronger conclusion if the authors specifically provided a set of concrete recommendations to increase language (and other kinds) of representativeness. A good place to start to look into such guidelines might be literature on community-based research and patient-centered clinical research (PCORI, a federally-funded organization does just this).

Trials that do not require English may already be exclusive "by default" - the next step would be looking at actual recruitment for those trials vs. the ones that are English-exclusive. I would imagine that the differences in the proportion of LEP subjects would be negligible and that the only significant differences would be in those trials that actively recruit non-English speakers.

It is also important to acknowledge that English proficiency requirements are only part of the story, it is equally important for clinical research to actively seek recruitment from populations that are underrepresented due to other socio-economic factors. The COVID-19 crisis underscores the importance of inclusivity in clinical research: in the US, rates of vaccination are much lower in black and Hispanic populations than in whites. In order to understand why this is the case and more urgently, begin to close those gaps, inclusive research is essential.

Finally, it may be somewhat out of the scope of this paper, but the lack of translation resources for LEP researchers in other settings (particularly low and middle-income countries) also means that research not conducted in English has few opportunities to be funded or published by major organizations. It may be worthwhile to mention that journals and funding agencies could do more to facilitate research with a global impact, which is particularly important given the global epidemic.

It is perhaps somewhat understandable that researchers simply do not have the time, training or capacity to conduct outreach efforts that include medically-underserved populations. However, without those efforts, clinical research will increase existing inequalities. Until clinical research makes a real effort to include underrepresented populations, we cannot expect medicine and healthcare to have equitable impact.

[LINK]

---

## [Decision Letter · Decision Letter 2]

15 Jul 2021

Dear Dr. Bierer,

Thank you very much for re-submitting your manuscript "Evaluating the frequency of English language requirements in clinical trial eligibility criteria: a systematic analysis using ClinicalTrials.gov" (PMEDICINE-D-21-01563R2) for review by PLOS Medicine.

I have discussed the paper with my colleagues and the academic editor and it was also seen again by three reviewers. I am pleased to say that provided the remaining editorial and production issues are dealt with we are planning to accept the paper for publication in the journal.

[LINK]

We look forward to receiving the revised manuscript by Jul 16 2021 11:59PM.   

Sincerely,

Beryne Odeny, 

Associate Editor 

PLOS Medicine

plosmedicine.org

Requests from Editors:

Thank you for revisions. Please address the following points before we proceed:

1) Please include p-values for all estimates provided in the results section - text and tables. For instance, tables 2 & 3 have 95% CIs but no p-values

2) References – please consistently include access dates for weblinks e.g., refs #1, #6, #9

Comments from Reviewers:

Reviewer #1: The authors have addressed my concerns and I now recommend publication.

Peter Flom

Reviewer #2: The revised paper provides thoughtful responses to the reviewer critiques, and has made edits and additions that strengthen the manuscript. My one additional comment on this read is that it would be helpful to know more about the characteristics of the 390 trials that specifically mentioned accommodation of translation to another language. If the reasons for affirmative accommodation of translation are related to geography, specific conditions likely to occur in non-English-speaking populations, or other identifiable correlates, this would be an important additional finding that would help readers to better understand how language requirements in clinical trials are being handled.

Reviewer #3: The authors did a good job of taking into account reviewers' comments and editing the manuscript accordingly. They likewise clearly explained their reasons when deciding not to change the manuscript.

I recommend this article for publication.

[LINK]

---

## [Decision Letter · Decision Letter 3]

5 Aug 2021

Dear Dr Bierer, 

On behalf of my colleagues and the Academic Editor, Dr. Margaret E Kruk, I am pleased to inform you that we have agreed to publish your manuscript "Evaluating the frequency of English language requirements in clinical trial eligibility criteria: a systematic analysis using ClinicalTrials.gov" (PMEDICINE-D-21-01563R3) in PLOS Medicine.

PRESS

Sincerely, 

Beryne Odeny 

Associate Editor 

PLOS Medicine